# Pathophysiological Mechanisms of Cardiac Dysfunction in Transgenic Mice with Viral Myocarditis

**DOI:** 10.3390/cells12040550

**Published:** 2023-02-08

**Authors:** Matthias Rohrbeck, Verena Hoerr, Ilaria Piccini, Boris Greber, Jan Sebastian Schulte, Sara-Sophie Hübner, Elena Jeworutzki, Carsten Theiss, Veronika Matschke, Jörg Stypmann, Andreas Unger, Huyen Tran Ho, Paul Disse, Nathalie Strutz-Seebohm, Cornelius Faber, Frank Ulrich Müller, Stephan Ludwig, Ursula Rescher, Wolfgang A. Linke, Karin Klingel, Karin Busch, Stefan Peischard, Guiscard Seebohm

**Affiliations:** 1Institute for Genetics of Heart Diseases (IfGH), Department of Cardiovascular Medicine, University Hospital Münster, D-48149 Münster, Germany; 2Translational Research Imaging Center, Clinic of Radiology, University Hospital Münster, D-48149 Münster, Germany; 3Human Stem Cell Pluripotency Laboratory, Max Planck Institute for Molecular Biomedicine, D-48149 Münster, Germany; 4Chemical Genomics Centre of the Max Planck Society, 44227 Dortmund, Germany; 5Institute of Pharmacology and Toxicology, University Hospital Münster, D-48149 Münster, Germany; 6Department of Cytology, Institute of Anatomy, Ruhr-University Bochum, D-44780 Bochum, Germany; 7Department of Cardiovascular Medicine, Division of Cardiology, University Clinic Münster, 48149 Münster, Germany; 8Institute of Physiology II, Faculty of Medicine, University of Münster, D-48149 Münster, Germany; 9Institute of Virology Münster (IVM), Centre for Molecular Biology of Inflammation (ZMBE), University of Münster, D-48149 Münster, Germany; 10Research Group Regulatory Mechanisms of Inflammation, Institute of Medical Biochemistry, Centre for Molecular Biology of Inflammation, University of Muenster, 48149 Muenster, Germany; 11Cardiopathology, Institute for Pathology and Neuropathology, University Hospital of Tübingen, D-72076 Tübingen, Germany; 12Institute of Integrative Cell Biology and Physiology, Faculty of Biology, University of Muenster, Schlossplatz 5, 48149 Muenster, Germany

**Keywords:** CVB3, myocarditis, contractility, cardiac dysfunction, calcium homeostasis

## Abstract

Viral myocarditis is pathologically associated with RNA viruses such as coxsackievirus B3 (CVB3), or more recently, with SARS-CoV-2, but despite intensive research, clinically proven treatment is limited. Here, by use of a transgenic mouse strain (TG) containing a CVB3ΔVP0 genome we unravel virus-mediated cardiac pathophysiological processes in vivo and in vitro. Cardiac function, pathologic ECG alterations, calcium homeostasis, intracellular organization and gene expression were significantly altered in transgenic mice. A marked alteration of mitochondrial structure and gene expression indicates mitochondrial impairment potentially contributing to cardiac contractile dysfunction. An extended picture on viral myocarditis emerges that may help to develop new treatment strategies and to counter cardiac failure.

## 1. Introduction

Most RNA viruses transmit their genetic information via single-stranded RNA and cause diseases such as the common cold, COVID-19, influenza, viral meningitis, encephalitis, heart failure, late onset diabetes (type1) and others. The financial burden by these viral diseases is tremendous.

One of these relevant viral pathogens is coxsackievirus B3 (CVB3), which belongs to the picornaviridae family and is a member of the Enterovirus genus. CVB3 causes meningitis [1,2] and pancreatitis [3,4], but most clinical relevance is attributed to pathogenesis of myocarditis in children and juveniles [5,6,7,8,9]. Out of the six different serotypes (CVB1–6), CVB3 is the predominant agent of viral myocarditis being responsible for 20–40% of acute heart infections. Furthermore, prolonged viral persistence potentially ends up in dilated cardiomyopathy (DCM) [10]. The clinical phenotype depends on unclear host factors and differs widely from unapparent inflammation and rapid viral clearance to acute congestive heart failure or viral persistence leading to chronic inflammation with progressive chronic disease [11,12].

Despite the increasing and profound knowledge regarding viral myocarditis, it is still a challenge to identify further host factors, which contribute to the development of chronic myocarditis. A solid diagnosis and treatment of patients demands a better understanding of the inflammatory process and the impact of the viral proteins on the host cell [13,14]. Murine models of CVB3-induced myocarditis help in gaining information about the interactions between the host cell and viral proteins, respectively, to determine the affected host factors. Most experimental approaches use susceptible mouse strains such as A.BY/SnJ mice or BALB/c H-2^d^ and provide unique insight into the acute infection phase or subacute phase of the inflammation [15,16]. CVB3 is a relevant example causing acute viral myocarditis and the viral life cycle of CVB3 has been extensively studied [17]. However, the development of chronic infection or the determining of host-signaling factors involved in the pathogenesis of DCM need to be explored further. Murine models mimicking enteroviral infection running a chronic course are accompanied by virus RNA persistence, limited viral plus-strand RNA synthesis and protein expression and chronic inflammation [18].

According to present knowledge, different mechanisms might be involved in progressive pathogenesis of viral myocarditis. For instance, the direct cell death of cardiomyocytes is induced by CVB3 due to cell lysis. Furthermore, an overshoot reaction of infiltrating immune cells is anticipated [17]. Cardiac remodeling due to prolonged chronic inflammation is caused by dysregulation of different matrix metalloproteinases (MMP) depending on the examined mouse strain [16,19,20,21]. In addition, an enhanced generation of reactive oxygen species (ROS) induced by CVB3 within the cytoplasm results in an increased permeability of the mitochondrial outer membrane and leads to a destabilization of the mitochondria with severe consequences to the physiology of cardiomyocytes [22]. Yet, there remains a missing link between pathogenesis, cardiac remodeling and contractile dysfunction.

Recently, we established an inducible CVB3-transgenic non-infectious human iPSC model [23]. In human iPSC-derived cardiomyocytes, we found reduced and irregular beating rates, impaired mitochondrial function as indicated by increased ROS-production and altered β–adrenergic repolarization response in four-week-maturated cardiomyocytes expressing the non-infectious CVB3 variant, CVB3DVP0, which lacks correct VP0 capsid protein precursor cleavage and so hinders the correct formation of the viral capsid [24].

Here, we present an analogous transgenic mouse model that allows homogenous virus protein expression in the whole heart mimicking chronic myocarditis leading to a mild DCM accompanied by decreased cardiac output and ejection fraction in vivo. We investigated the active and passive features of cardiac CVB3ΔVP0 expression and characterized this novel transgenic model in vivo and in vitro. Our data provide prominent evidence for exclusive, CVB3-protein expression-based regulation of mitochondrial degeneration and dysfunction. Associated pathophysiological heart parameters resulted from a decreased contractile force and the disruption of excitation-contraction coupling especially in the late phase of contraction under stress as a consequence of an impaired calcium homeostasis. Together, our data show how the viral protein expression in cardiomyocytes determines the outcome of chronic viral myocarditis in vivo and in vitro.

## 2. Materials and Methods

### 2.1. Transgenic CVB3ΔVP0 Mice

Generation of heterozygous transgenic CVB3ΔVP0 mice (TG) has been reported previously [25]. Breeding was performed in accordance with German guidelines. Housing and animal experiments were approved by the Landesamt für Natur, Umwelt und Verbraucherschutz NRW and were performed in accordance with the guidelines of the Westfälische Wilhelms-University Münster. Unless not otherwise quoted, all described experiments were performed on male mice at 25 ± 2 weeks of age and the genotype of every animal was confirmed by polymerase chain reaction (PCR). All mice were on C57Bl/6 J background and littermates or imported mice (Charles River) were used as wildtype (WT) controls. All animals were provided food and water ad libitum.

### 2.2. Cardiac Magnetic-Resonance-Imaging (CMR)

In vivo MR imaging was performed at 9.4 T on a Bruker BioSpec 94/20 (Ettlingen, Germany) equipped with a 1 T/m gradient system and ParaVision 5.1 operating software including the IntraGate software for sequence acquisition and reconstruction. A 35 mm volume coil was used for data acquisition. To measure the systolic heart function, a stack of contiguous short-axis slices was acquired to cover the entire right and left ventricles by using the self-gated cine FLASH (IntraGate FLASH, Bruker, Ettlingen, Germany) sequence with the following parameters: slice thickness, 1 mm; number of slices, 9; matrix size, 232 × 232; field of view, 30 × 30 mm^2^; averages, 1; TE, 3 ms; flip angle, 15°; pulse shape, Gaussian; pulse length, 0.5 ms; TR, 7.4 ms; number of repetitions, 100; acquisition bandwidth, 75 kHz and scan time per slice, 2 min 54 s. The diastolic function was measured by using the self-gated cine UTE (IntraGate UTE) sequence with the following parameters: slice thickness, 1 mm; number of slices, 9; matrix size, 156 × 156; field of view, 20 × 20 mm^2^; averages, 1; TE, 0.314 ms; TR, 6.2 ms; flip angle, 15°; pulse shape, Gaussian; pulse length, 0.3 ms; number of projections, 246; polar undersampling, 2.0; number of movie cycles, 1000; effective time resolution, 2 ms (50–70 frames/cardiac cycle); acquisition bandwidth, 100 kHz and scan time per slice, 25 min 25 s.

During MRI measurements animals were anesthetized with isoflurane (1.5–2.5 vol% isoflurane, 0.7/0.3 air/O_2_ mixture) and were monitored for core body temperature and respiration rate using an MRI compatible monitoring system (SA Instruments, Stony Brook, NY, USA).

### 2.3. CMR Analysis

Volumetric analysis of the left and right myocardium and ventricles was performed using Amira (version 5.4.0, Visage Imaging GmbH, Berlin, Germany) and Segment (version v1.9, Medviso AB, Lund, Sweden) software. To determine the systolic function, the regions of interest in the MR FLASH images were selected manually on the end-diastolic and end-systolic frame of each slice by tracing the epicardial and endocardial borders. In order to obtain the global parameters of the entire heart the volume for each frame was calculated as the sum of the area of interest in each slice multiplied by the slice thickness. Stroke volume, cardiac output per minute and the ejection faction were calculated from the blood volume, determined in the end-systolic and end-diastolic phase. Left ventricular (LV) and RV mass was calculated by multiplying the volume of the LV and RV myocardium by the tissue density of 1.05 g/cm^3^. To analyze diastolic function, a central slice in short axis view of the left ventricle was segmented in high temporal resolution using the segment. The resulting volume–time curve was fitted with a spline function and the corresponding first derivative calculated, providing the early (E) and late (A) filling rate.

#### 2.3.1. Doppler-Echocardiography

Cardiac Doppler echocardiography was performed on a specialized ultrasound system (VEVO 2100; VisualSonics Inc., Ontario, Canada). Following induction of anesthesia with isoflurane (2.0% isoflurane, 98% O_2_), examinations were performed with the system’s 40 MHz linear probe. ECG-triggered parasternal short-axis views were obtained together with a parasternal long-axis view. Routinely pulsed-wave Doppler signals of the velocity time integral of the mitral valve inflow, the aortic outflow and the pulmonic outflow were obtained. Ultrasound image analyses were performed according to the American Society of Echocardiography’s (ASE) leading edge method (using the VEVO 2100 implemented image analyses software.

#### 2.3.2. Electrocardiographic Recordings

Mice were anaesthetized with 1.5% isoflurane in a mixture of gases containing 33% oxygen and 66% nitrous oxide and spontaneously breathing while body temperature was retained between 37 and 38 °C. Sufficient depth of anesthesia was confirmed by frequent toe pinching. Recordings of six-lead surface ECG were performed using subcutaneous needle electrodes. Signals were recorded unfiltered with scanning rate of 1 k/s (1 kHz). After a baseline ECG for 5 min a β-adrenergic stimulation was achieved by intraperitoneal injection of isoproterenol (2 mg kg^−1^ body weight; Sigma-Aldrich) dissolved in water and diluted in saline solution. After drug application the ECG recordings were continued and analyzed for an additional 20 min. Data analysis was performed with LabChart Pro 8.0.9 (ADinstruments, Dunedin, New Zealand) and the signal was digitally filtered using a band-pass filter type with high cut-off frequencies of 500 Hz and low cut-off frequencies of 15 Hz. In a timeframe of 15 s, R waves >150 consecutive heartbeats were aligned to average signal ECGs and the results were analyzed and revised manually using LabChart Pro 8.0.9.

#### 2.3.3. Isolation of Ventricular Cardiomyocytes

Sedation and euthanasia were performed by intraperitoneal injection of ketamine (65 mg/kg bodyweight) and xylazine (13 mg/kg bodyweight). Following thoracotomy, the beating heart was excised from the thoracic cavity and the exposed aorta was cannulated in a timely manner. The extracted hearts were fixed to a Langendorff apparatus and perfused for 5 min at a flow velocity of 2.5 mL/min with prewarmed Ca^2+^-free buffer (buffer A) composed of (in mM) 133.47 NaCl, 3.99 KCl, 1.37 NaH_2_PO_4_ × 1 H_2_O, 1.0 MgCl_2_ × 6 H_2_O, 10 HEPES and 10 D(+)-glucose × 1 H_2_O (pH 7.47 adjusted with NaOH) additionally containing heparin (14.3 IE/mL heparin-sodium) and 10 mM ATPase-inhibitor 2,3-butanedione-monoxime (BDM). The enzymatic digestion was performed for 6 min with buffer A supplemented with 0.8 mg/mL collagenase (230 U/mg CLS-2 Typ II, Worthington, Lakewood, NJ, USA) and 12.5 µM CaCl_2_. Finally, the digested ventricles were dissected into several small pieces and separated by gravity flow through a nylon mesh into ventricular myocytes.

### 2.4. Patch-Clamp Technique

Dissociated ventricular cardiomyocytes were electrophysically studied using whole-cell configuration of the patch-clamp technique. Furthermore, 200 µL cell suspension was added to 2 mL extracellular solution in an experimental glass-bottom dish mounted on an inverted microscope (Axiovert 200, Carl Zeiss, Cambridge, United Kingdom). The extracellular solution containing (in mM): 136 NaCl, 5.4 KCl, 0.33 NaH_2_PO_4_ × 1 H_2_O, 1.0 MgCl_2_ × 6 H_2_O, 10 HEPES, 10 D(+)-glucose × 1 H_2_O and 1 CaCl_2_ × 1 H_2_O adjusted to pH 7.4 with NaOH. Borosilicate glass capillaries, GB150TF-8P (Harvard Apparatus, Massachusetts, MA, USA), were pulled with a horizontal P-97 Micropipette Puller (Sutter Instruments, Novato, CA, USA) and were used as patch pipettes. Tip resistance was 1.5–3.5 MΩ. The pipette was filled with intracellular solution containing (in mM): 110 CsCl, 30 TEA-Cl, 10 NaCl, 10 HEPES and 5 Mg-ATP adjusted to pH 7.2 with CsOH. For data acquisition, an EPC 10 USB Amplifier (HEKA, Lambrecht, Germany) was connected to a computer using PatchMaster Software Version 1.2. (HEKA). For recording of action potentials (AP), a high resistance seal was established and finally the ruptured patch clamp method was used. After leak current compensation, the cells were stimulated successively with increasing frequencies starting from 0.5 to 10 Hz, respectively, for 1 min. After achieving Ca^2+^-overload, one action potential was triggered for early detection after depolarizations or delayed after depolarizations.

### 2.5. Cellular Ca^2+^ Transient [Ca]_I_/Calcium Imaging/Shortening

Ventricular cardiomyocytes were incubated for 10 min with the fluorescent Ca^2+^ indicator Indo-1 (9 µM; Indo-1/AM, Molecular Probes, Thermo Fisher Scientific, Waltham, MA, USA) at room temperature. Indo-1-loaded cells were transferred to a recording chamber filled with Tyrode’s solution (containing in mM: 140 NaCl, 5.8 KCl, 0.5 KH_2_PO_4_, 0.4 Na_2_HPO_4_ × 2 H_2_O, 0.9 MgSO_4_ × 7 H_2_O, 10 HEPES, 10 glucose, 2 CaCl_2_, pH 7.3 adjusted with NaOH) and mounted on the stage of an inverted microscope (Eclipse Ti-S, Nikon, Tokyo, Japan). After allowing cells to settle down for 5 min and washing out excess dye by perfusion with Tyrode’s solution, cardiomyocytes were field-stimulated for 10 min at 0.5 Hz (Myopacer, Ionoptix, Milton, MA, USA) before recording started. Intracellular Ca^2+^ transients and sarcomere shortening were recorded simultaneously (Myocyte Calcium and Contractility System, Ionoptix, Milton, MA, USA). Indo-1 was excited at 340 nm by a continuous light source (UVICO, Rapp OptoElectronic, Wedel, Germany) equipped with respective low- and bandpass filters. The emitted fluorescence (405 and 495 nm) was detected by photomultipliers. Sarcomere shortening was captured with a CCD camera. Data recording and analysis was performed with Ionwizard software Version 6.3. (Ionoptix, Milton, MA, USA). Transients from at least 10 cardiomyocytes were recorded under basal conditions before cells were superfused with Tyrodes’s solution containing 10^−6^ M isoproterenol. After reaching steady-state conditions, another 10 cardiomyocytes were recorded under exposure to isoproterenol.

### 2.6. Light Microscopy

After the isolation procedure, ventricular cardiomyocytes were transferred into 35 × 10 mm^2^ cell culture dishes, settled for 5 min at RT and examined for quality and quantity yield of the isolation. Cell areas and lengths were detected using a microscope (Eclipse Ti-S, Nikon, Tokyo, Japan) with 20× DIC M N2 objective and connected CoolSNAP HQ2 camera.

### 2.7. Electron Microscopy

Mice were anesthetized and perfused transcardially with PBS for 60 s followed by 2.5% (*v*/*v*) glutaraldehyde (GA) for electron microscopy. The heart was dissected en bloc with the aid of a binocular microscope, avoiding any traction of the tissues. After post fixation for 1 day in GA, the tissue was washed in PBS for 10 min. Specimens were then incubated with Dalton for 2 h followed by washing in PBS for 10 min. After dehydration in an ascending ethanol series, the tissue was embedded in Epon, at ratios of propylene oxide and Epon, first 3:1 for 45 min, 1:1 for 1 h and then 1:3 for 2 h and as the last step, pure Epon overnight. Sections were cut with an Ultracut E Reichert-Jung, collected on formvar-coated grids and contrasted with uranyl acetate. Samples were analyzed with a Philips EM 420 (Philips, Holland) transmission electron microscope equipped with a digital CCD camera (Model 792 BioScan; Gatan, USA) and photographic plates system processed with a Ditabis Micron System. Measurements were performed with the aid of ImageJ Version 1.49. The collected data were analyzed with Statistica Version 12.7, by StatSoft. A *t*-test for independent sampling was performed.

### 2.8. Genome-Wide Expression Analysis

Isolated ventricular cardiomyocytes (isolated as described above) from three WT and two CVB3ΔVPO mice were shock-frozen in liquid nitrogen and stored at −80 °C till experimental usage. Total RNA was extracted using NucleoSpin RNA (Macherey-Nagel; including on-column DNAse treatment) and quantified by UV-spectrophotometry. Labelled cRNA was prepared from 250 ng DNA-free RNA samples using TotalPrep linear RNA amplification kits (Life Technologies #AMIL1791) and validated for integrity by capillary electrophoresis using an Agilent 2100 Bioanalyzer (Agilent technologies, Boeblingen, Germany). Microarray hybridizations on MouseWG-6 v2.0 Expression BeadChip (6 samples per BeadChip, >45,000 mouse targets per sample) were performed following manufacturer´s recommendations. Cy3-stained chips were scanned using HiScan SQ instrumentation. Acquired data were processed using GenomeStudio software Version 2.0.5. Differential expression analysis was performed between CVB3ΔVPO and WT samples after background subtraction using cubic spline normalization and Illumina Custom error model without FDR. Transcripts with expression fold change ≥2 and significant *p*-values of ≤0.01 were considered significantly dysregulated in CVB3ΔVPO ventricular cardiomyocytes compared to WT.

### 2.9. RT-qPCR

The mRNA expression level results obtained from genome-wide expression analysis were validated by using real-time polymerase chain reaction (RT-qPCR). Total RNA from ventricular cardiomyocytes from additional 5 WT and 5 CVB3ΔVPO mice was isolated as described above. Furthermore, 350 ng total RNA was reverse-transcribed into cDNA using oligo-dT_15_ primers and M-MLV reverse transcriptase (Affymetrix, Santa Clara, CA, USA) at 42 °C. All primer pairs were designed with Primer3 software 4 February 2015 (http://primer3.ut.ee/, accessed on 4 February 2015) and validated with respect to efficiency and specificity. Primers used are listed in Table 1.

Experiments were conducted using iTaq Universal SYBR Green Supermix (Biorad) on an Applied Biosystems 7900HT Fast Real-Time PCR System (Life Technologies, Darmstadt, German). All measurements were performed in duplicate. Relative quantification of expression was performed according to the “ΔΔCt method” [26] by employing beta-actin (*Actb*) for normalization. Two-sided Student’s *t* tests were performed on the basis of ΔCt values from biological replicates.

## 3. Statistics

Data shown are expressed as mean ± standard error of the mean (SEM). Statistical analyses were performed with Prism 6.01 (GraphPad Software, Inc., La Jolla, CA, USA) for direct comparison between both genotypes (wildtype versus transgenic) using Mann–Whitney U-test. Calculated *p*-values using Mann–Whitney U-test with *p*-value < 0.05 were considered to be statistically significant.

## 4. Results

### 4.1. CVB3 Mice Exert Impaired Cardiac Performance

A mouse strain expressing the viral genome CVBΔVP0 exclusively in cardiac cells was raised as described before [25,27]. The genome contained two point mutations preventing auto-proteolysis of VP4-VP2 and thus formation of functional viral capsids rendering the system non-infectious (Figure 1a) [25,27]. However, in transgenic CVBΔVP0 cells, viral proteins, including cytotoxic proteases, are expressed [25]. Macroscopic inspection of 25-week-old CVB3 mice and wild type littermates (control) showed no obvious differences in appearance. Anaesthetized 25-week-old transgenic (TG) CVB3 and control mice were subjected to magnetic resonance imaging (MRI) in order to determine cardiovascular parameters (Figure 1b,c): Myocardial mass of right and left ventricles were measured and showed no abnormalities. End-diastolic volume of the left and right ventricle were without any statistically significant pathological findings as well (Appendix A). However, the end-systolic volume showed increased values due to reduced stroke volume, cardiac output and ejection fraction of the left and right ventricle (Figure 1c). Even though the ejection fraction in CVB3-expressing mice (mean 61%) was significantly reduced compared to the control group (mean 70%), the heart function was retained when considering a critical threshold of heart insufficiency of 40–50%. Analyses of diastolic function marked CVB3-impact on late diastole: While the early filling rate (E-peak) was preserved, CVB3 mice showed a largely reduced late atrial ejection (A-peak). As a consequence, the E/A ratio was substantially increased in transgenic mice (1.99) compared to the controls (1.59) (Figure 1d,e).

In tissue Doppler imaging (TDI) additional information about the E’ and A’ tissue velocities were obtained. CVB3 mice showed similar A’-velocities compared to wt littermates. However, E’-velocities were significantly decreased and thus corresponding E’/A’ ratios (Figure 1f). Taken together, the functional parameters describe a disordered cardiac function, which indicates a degree 2 diastolic dysfunction and thus characterize the clinical picture of HFpEF (heart failure with preserved ejection fraction).

### 4.2. Marked Alterations in ECGs of CVB3 Mice

Electrocardiograms (ECGs) were studied in anaesthetized wt and transgenic CVB3 mice. Anaesthetized transgenic mice showed marked and frequent ECG alterations, which were rarely seen in wt mice (examples of which seen in Figure 2a). These alternates refer to P/T-wave and arrhythmic events (Figure 2b) and rendered QTC-analyses difficult. Still, a tendency towards elongated QT intervals by about 3 ms can be averaged. As β-adrenergic signaling is a key modulator of the physiologic cardiac stress reaction and is a critical parameter in chronic myocarditis in patients, β-adrenergic signaling was stimulated by isoprenaline injections. Upon acute injection, the QT intervals transiently normalized, but were elongated 10 min after isoprenaline injection (Figure 2c).

### 4.3. Altered Intracellular Calcium Handling in Murine CVB3 Cardiomyocytes

Altered ventricular functions seen in MRI, US and ECGs suggest that cardiac myocytes may be affected by CVB3-expression. Isolated ventricular CVB3-expressing cardiomyocytes revealed a mild increase in length and area compared to wt cells consistent with impaired intracellular structures (Figure 3a,b, Appendix A). For further functional analysis of isolated cardiomyocytes, action potentials were recorded via patch clamp. Action potential length under resting conditions showed no prominent alteration of action potential duration under rest. Consistent with unaltered action potentials at basal conditions, no obvious changes in the respective expression patterns of L-type calcium channels (Ca_V_1.2), ryanodine receptors (RYR2a), K_V_7.1 and hERG were found in control versus CVB3-expressing cardiomyocytes (CVB3-VP1 and ion channel immunostainings are shown and quantified in Appendix A). To assess Ca^2+^-overload-triggered arrhythmias, cardiomyocytes were paced at 0.5–7 Hz. At higher frequencies, which induce Ca^2+^-overload, CVB3 cells especially tended to develop delayed afterdepolarizations (Figure 3c,d). Reduced end-diastolic function and tendencies to develop arrhythmias under β-adrenergic stimulation in whole animals and by high pacing frequencies in single cells indicate dysregulated intracellular calcium handling. In order to characterize intracellular Ca^2+^-handling, Ca^2+^-transients and sarcomere shortening were carefully recorded simultaneously (Figure 4a). Ca^2+^_i_-release was heterogeneous in CVB3-expressing myocytes and on average delayed compared to WT myocytes under basal conditions (Figure 4b). The Ca^2+^-transient decay and sarcomere relaxation were both retarded as well under basal conditions as indicated by increased tau values for both parameters (Figure 4d). This indicates a disturbed intracellular Ca^2+^ handling already under basal conditions.

β-adrenergic stimulation is the main physiological mechanism to increase cardiac function in phases when higher cardiac performance is required. To investigate the effect of β-adrenergic stimulation on intracellular Ca^2+^-handling, the cells were stimulated with isoproterenol. While Ca^2+^_i_-release was normalized to wt levels in CVB3 myocytes, the Ca^2+^-transient decay was still delayed compared to wt cells (Figure 4b,d). In contrast, sarcomeric relaxation was no longer increased in CVB3-expressing cardiomyocytes. Thus, Ca^2+^ uptake was uncoupled from sarcomere relaxation under β-adrenergic stimulation. Moreover, the Ca^2+^-transient amplitudes were not different between genotypes under basal and isoproterenol-stimulated conditions, but sarcomere shortening did not reach the same amplitude in CVB3 myocytes compared to wt under β-adrenergic stimulation (Figure 4c). A generally slowed Ca^2+^ reuptake (Figure 4d), which could promote Ca^2+^ overload and arrhythmia, and reduced maximal sarcomere shortening under β-adrenergic stimulation, mimics the phenotype in patients prone to decompensation during stress situations [28].

### 4.4. Ultrastructural Analyses Suggest Mitochondrial Degradation as a Result of Altered Mitochondrial Gene Expression

Application of electron microscopy identified abnormalities in myofibrils as well as mitochondrial distribution and organelle structure within the cardiomyocytes of transgenic (TG) CVB3-expressing mice compared to wt controls. The subcellular cardiomyocyte structure was observed and analyzed. In control cells, numerous highly organized interfibrillar mitochondria were located alongside densely packed myofibrils, whereas the distribution, shape and size of mitochondria was altered in CVB3ΔVP0-expressing cells (Figure 5). Quantitative analysis revealed that the area of mitochondria was significantly decreased in CVB3ΔVP0-expressing cardiomyocytes (*n* = 38; 452+/−203 µm^2^) compared to wt controls (*n* = 38; 616+/×206 µm^2^) (Figure 5).

Gene array analyses were performed on isolated ventricular myocytes of wt and CVB3-expressing mice. The analysis of 45,000 genes identified 702 significantly altered genes with a cutoff at 2-fold change. This points towards a very restricted gene regulation by CVB3 of only 1.56% of the analyzed genes. The heatmap illustration clearly shows extensive transcriptional downregulation of 690/702 regulated genes, i.e., a total amount of 98.29% (Figure 6a). The gene ontology (GO) analysis of the gene array data clearly identified downregulation of mitochondrial host genes in transgenic CVB3 cardiomyocytes as the most prominent effect (Figure 6b). Further, genes relevant for phosphorylation seemed to be transcriptionally reduced. Using quantitative PCR, significant downregulation of Ndufb6, Atp5K and PPa2, i.e., genes critical for normal mitochondrial NAD^+^ or ATP production in complex I or V, were confirmed. Together, these data suggest that not only mitochondrial morphology, but also their function may be hampered in transgenic CVB3-expressing cardiomyocytes.

### 4.5. Sarcomeric Degradation of CVB3ΔVP0-Expressing Cardiomyocytes

The GO-analysis of the gene array data additionally suggests sarcomeric degradation in CVB3-expressing cardiomyocytes. Consistently, a disruption of cellular structures was detected in these cells. In some regions the sarcomeres were heavily disintegrated and shortened (Figure 5). However, the intact sarcomeric regions showed normal t-tubule distances (Appendix A). Control cells showed a tight coupling of calcium and contraction signals during basal measurement, while longer lasting, elevated calcium signals were detected during the late diastolic phase of the contraction in CVB3-expressing cardiomyocytes (Figure 4). Under β-adrenergic stimulation, the signals of calcium and contraction in control cells remained tightly coupled, while progressive calcium-contraction uncoupling in the diastolic phase was observed in CVB3ΔVP0 cardiomyocytes (Figure 4).

## 5. Discussion

Persistent CVB3 infection represents a relatively common causative or modulatory basis of chronic myocarditis. CVB3 infections cause a variety of host reactions including immune cell responses and fibrosis in the heart of patients as well as of mice. Acute CVB infection is known to induce lysis of cardiac myocytes. In order to study persistent CVB3 infection we established a transgenic mouse model with CVB3ΔVP0 expression limited to cardiomyocytes. This model expresses the CVB3ΔVP0 variant which generates viral proteins but cannot form capsids due to restricted VP0 precursor protein cleavage and is therefore considered non-infectious [25]. This unique mouse model mimics a uniformly infected heart with regard to CVB3 protein expression in cardiomyocytes, which allowed screening for functional and mechanistic effects in the whole heart (in vivo) and also in single cardiomyocytes (in vitro).

In this study, we found markedly reduced cardiac performance in transgenic CVB3-expressing hearts. Despite normal heart mass and normal ejection fraction, stroke volume and cardiac output were reduced suggesting that force generation in the late-phase diastole is reduced (Figure 1b,c). The end-systolic blood volume remaining in the ventricles was increased indicating that cardiac elasticity is reduced. Thus, marked changes in cardiac performance are associated with expression of CVB3 proteins in murine cardiomyocytes, which requires further investigation.

Although reduced function of atria was observed here, we concentrated on the ventricles that are responsible for about 85% of cardiac performance. Isolated relaxed ventricular myocytes expressing CVB3 proteins are slightly elongated, supposedly in part resulting from reduced gene expression of whole titin, a key component of cardiomyocyte elasticity (Figure 3a,b). Reduced cardiomyocyte elasticity will reduce whole heart elasticity and might cause increased end-systolic volume and limit the overall cardiac performance in animals (see Figure 1b–d). In ECG recordings a mild QTc interval extension was observed (Figure 2c). This effect could be transiently compensated by β-adrenergic stimulation via isoprenaline application. This re-occurrence of QTc interval extension suggests that transgenic CVB3-expressing animals underwent decompensation. Beside its transiently beneficial effect, β-adrenergic activation triggered distinct changes in ECG morphology and lethal arrhythmias predominantly in transgenic CVB3 protein-expressing mice (Figure 2a–c). The common physiological acute stress reaction in the heart is mediated by adrenergic signaling and the investigated CVB3 animal model mirrors the clinical situation well, as chronic myocarditis patients are often clinically stable in relaxed situations but often suffer life threatening decompensation under stress situations [29]. Single cell electrophysiology suggests that CVB3-protein expression sensitized cardiomyocytes for Ca^2+^-overload, a pivotal factor in arrhythmogenesis (Figure 3). Increased (intra-)cellular Ca^2+^-flux is one of the major effects of β-adrenergic stimulation and stimulated cytosolic Ca^2+^-release from the sarcoplasmic reticulum induces facilitated contraction (positive inotropic effects). However, the increased cytosolic Ca^2+^ has to be efficiently removed from the cytosol into the intracellular store units formed by the sarcoplasmic reticulum and to a lesser extent the mitochondria. The Ca^2+^-contraction coupling is highly controlled in healthy cardiomyocytes but uncoupled in CVB3ΔVP0 cardiomyocytes, supposedly due to insufficient Ca^2+^-reuptake (Figure 4b–d). Both, normal action potential morphology at rest and inconspicuous immune staining’s of ion channels involved in cardiac repolarizations, suggest no obvious alterations in ion channels in CVB3ΔVP0 cardiomyocytes (Figure 3, Appendix A). Thus, Ca^2+^-overload due to an unknown mechanism probably is the pro-arrhythmic basis of stress triggered arrhythmias under chronic viral heart infection (Figure 2, Figure 3 and Figure 4).

Cellular phenotype and subcellular structure were analyzed via electron microscopy. Cell length and area were significantly increased in CVB3ΔVP0-expressing cells (Figure 3a,b), whereas mitochondrial area was significantly decreased and sarcomeric structure appeared heavily de-organized (Figure 5).

In order to identify the mechanism underlying these phenomena, gene array analyses were conducted. Heatmap illustration shows a clear tendency of transcriptional downregulation in CVB3ΔVP0-expressing cells compared to control cells (Figure 6a). GO biological process and GO cellular components analyses identified y genes involved in mitochondrial function and structure to be predominantly downregulated (Figure 6b). Mitochondria are the power plants of the cells including cardiac myocytes. Downregulation of Ndufb6, ATP5K and PPA2, that are key to normal mitochondrial complex I (Ndufb6-NADH dehydrogenase complex) and complex V (PPA2-cytochrome C oxidase complex, F-type ATPase-phosphorylation by ATP5K) function, were confirmed by RT-PCR (Figure 6c). Further, CamKIδ, which is involved in mitochondrial gene reprogramming, was found downregulated in CVB3ΔVP0 cardiomyocytes (Figure 6c) [30]. CamKIδ regulation may represent an important mechanism contributing to mitotoxicity in CVB3ΔVP0 cardiac myocytes, but this still has to be further investigated. Electron microscopy (EM) uncovered a clear reduction of mitochondrial size, strongly supporting effects of CVB3ΔVP0 on mitochondrial structure (Figure 5).

Recently, it was shown that the mitochondrial respiratory chain has a critical antiviral role in acute CVB3 infection [31,32]. Downregulation of energy metabolism may represent a host response effective for acute virus elimination as effective virus replication is highly ATP-dependent [33,34,35]. In the cardiac system, ATP is crucial for two key processes in excitation–contraction coupling, the ATP-driven Ca^2+^ reuptake into the sarcoplasmic reticulum and the ATP-driven contraction crossbridge cycle. Reuptake of Ca^2+^ is mediated by ATP-driven sarcoplasmic/endoplasmic reticulum calcium ATPase (SERCA). During a contraction crossbridge cycle in myocytes, one ATP is required to form the bridge of myosin to the actin filament allowing for the power stroke and a second ATP to allow for the detachment of myosin. The whole cycle is initiated by binding of Ca^2+^ to troponin C on the actin filaments. As mitochondrial ATP synthesis seems to be reduced by the host cell to downregulate viral replication, negative side effects on calcium-contraction coupling and ATP-dependent Ca^2+^ uptake into the SR are expected. Thus, the observed features including stress-dependent decompensation seen in mice closely mimic the clinical situation.

In summary, in CVB3 protein-expressing cardiomyocytes, contractile elements and mitochondria are deregulated, which compromises excitation–contraction coupling in late phase action potentials. As a result, cardiac performance is reduced and stress-triggered decompensations with arrhythmic tendency evolve.

## Figures and Tables

**Figure 1 cells-12-00550-f001:**
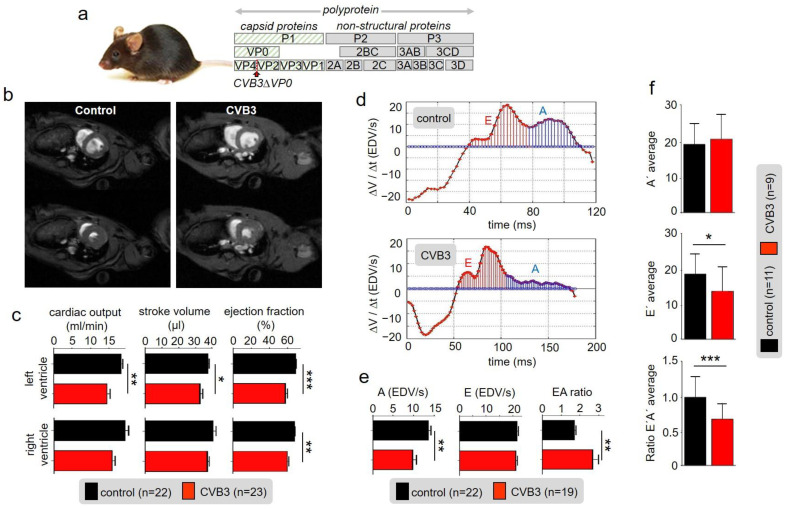
(**a**) Schematic illustration of the CVBΔVP0 expression system which was used to generate a transgenic mouse line. (**b**) Exemplary cardiac magnetic-resonance-imaging (CMR) of a wt mouse (left) and a CVB3ΔVP0 transgenic mouse (right). (**c**) Statistical analysis of measured cardiac output, ejection fraction and stroke volume (* indicates *n* < 0.05; ** indicates *n* < 0.01; *** indicates *n* < 0.001). (**d**) Volume–time filling curves of the left ventricle. (**e**) Statistical analysis of the early filling rate (E), atrial ejection (A) and EA ratio of CVB3 mice and control mice. (**f**) Analysis of tissue Doppler imaging (TDI) of transgenic CVB3 mice and control mice.

**Figure 2 cells-12-00550-f002:**
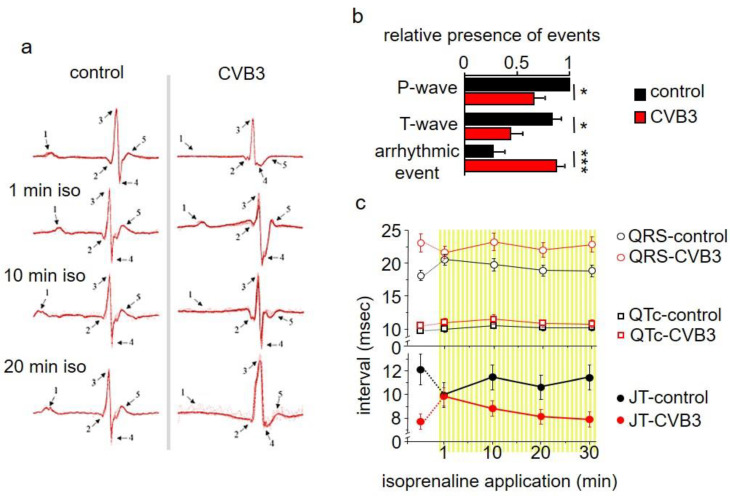
(**a**) Electrocardiogram (ECG) recordings of transgenic CVB3 mice and control mice under basal conditions and under β-adrenergic stimulation with isoprenaline. (**b**) Analysis of ECG recordings of CVB3 mice and control mice considering P-wave, T-wave and arrhythmic event emergence (* indicates *n* < 0.05; *** indicates *n* < 0.001). (**c**) Analysis of QT, QRS and JT (JT = QT − QRS duration) of transgenic CVB3 mice and control mice under basal conditions and after isoprenaline application.

**Figure 3 cells-12-00550-f003:**
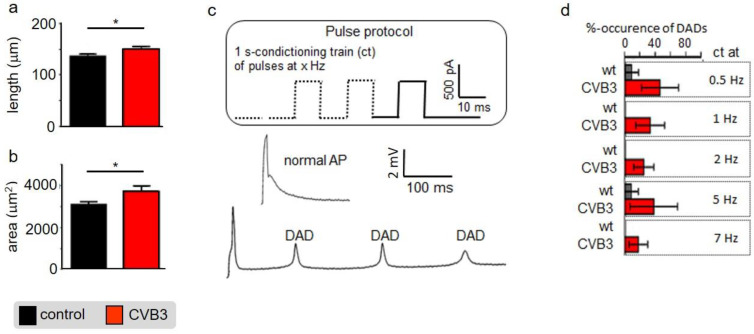
(**a**) Isolated mouse cardiomyocytes were optically measured under control conditions and under CVB3 expression. The cell length was quantified (n = 11; * indicates *n* < 0.05;) (**b**) Isolated mouse cardiomyocytes were optically measured under control conditions and under CVB3 expression. The cell area was quantified (n = 11; * indicates *n* < 0.05;) (**c**) Two exemplary induced action potentials of two different cardiomyocytes during patch clamp recordings and statistics on occurrence of delayed action potentials as relative events after stimulation at 0.5 to 7 Hz. (**a**) Example of a normal and a train of delayed depolarizations (DAD). (**d**) Cardiomyocytes isolated from transgenic CVB3ΔVP0 mice show a higher number of DAD events than the control group. The mean values ± SEM and the statistically relevant differences after a Mann–Whitney test (non-parametric) were plotted (*n* = 10–12 each group).

**Figure 4 cells-12-00550-f004:**
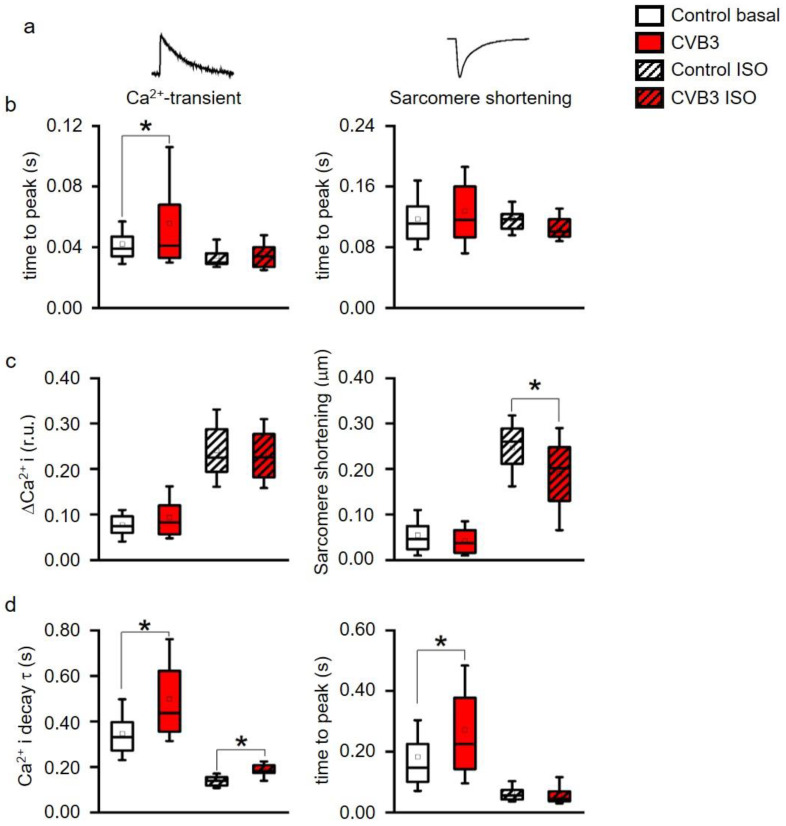
(**a**) Time to peak of intracellular Ca^2+^ (Ca^2+^_I_) transients and of sarcomere shortening, (**b**) Ca^2+^-transient (ΔCa^2+^_i_) and sarcomere shortening amplitudes, (**c**) time constant τ of a single exponential decay function fitted to the decay phase of Ca^2+^-transients and to sarcomere relaxation phase. * indicates *p* < 0.05 vs. WT controls. All parameters recorded under acute stimulation with isoproterenol (ISO) were significantly altered vs. basal (*p* < 0.05, not indicated in the figure for clarity). (box: 25th–75th percentile, whiskers: 10th–90th percentile, horizontal line: median, square: mean). (**d**) decay of Ca^2+^ and time to peak were simultaneously measured and analyzed. Ca^2+^ decay time was significantly altered between control and CVB3-expressing cardiomyocytes under basal and stimulated conditions. Time to peak was only altered under basal conditions indicating a calcium-contraction uncoupling due to CVB3 expression (* indicates *p* < 0.05).

**Figure 5 cells-12-00550-f005:**
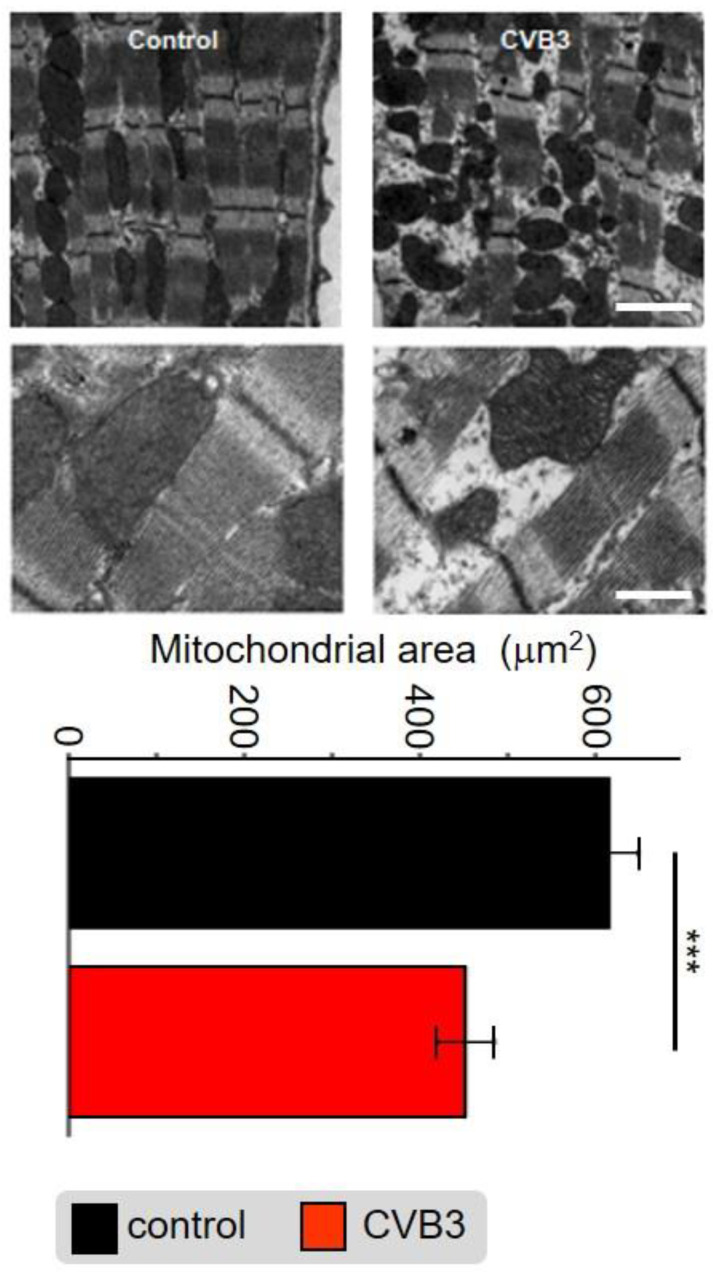
Electron microscopic image of sarcomeric structures with co-localized mitochondria of mouse cardiomyocytes from wt mice (left) and CVB3ΔVP0-expressing mice (right). Scale bar top right—100 nm, scale bar bottom right—30 nm. Statistical analysis of the mitochondrial area in mouse cardiomyocytes from wt mice and CVB3ΔVP0 expressing mice shows mitochondrial degradation due to CVB3 expression (*** indicates *p* < 0.001; *n* = 5).

**Figure 6 cells-12-00550-f006:**
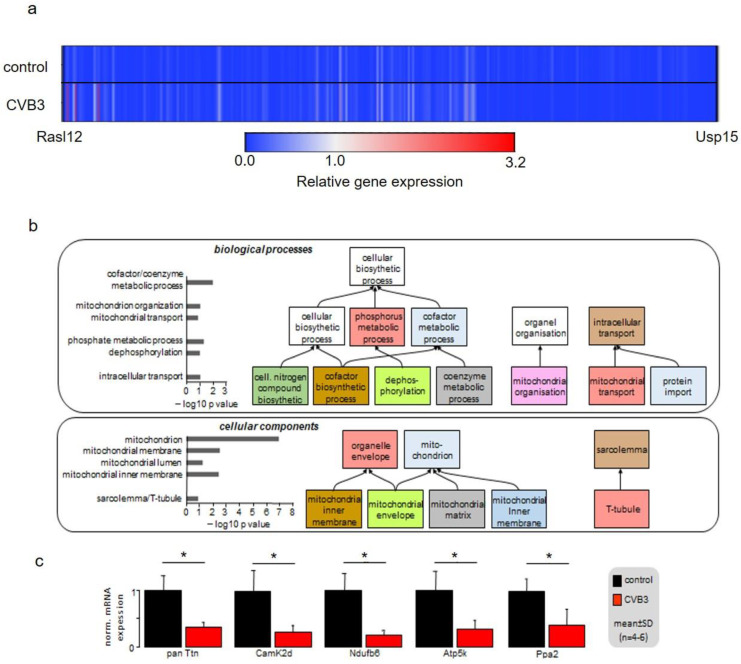
(**a**) Gene expression pattern of control and CVB3-expressing mice illustrated as a heatmap. Gene expression levels were averaged and normalized to the expression of cardiac ACTC1, which were not altered under CVB3 expression. (**b**) Gene array analysis and qPCR validation of regulated key genes in age-matched wt and CVB3-transgen mice. GO analyses of biological processes and cellular components are shown. (**c**) RT-PCR verified downregulation of five key genes (* indicates *p* < 0.05).

**Table 1 cells-12-00550-t001:** Used primers for validation of distinct results obtained from genome-wide expression analysis by RT-qPCR.

Gene	Product Size (bp)	Fwd Primer (5′-3′)	Rev Primer (5′-3′)	Remarks
*Actb*	78	ACTGCCGCATCCTCTTCCTC	ACATCTGCTGGAAGGTGGACA	CDS
*Ndufb6*	71	GGTCCTTCAGCCATCGTCTCCTTAG	AGGGTACACGCCGGATGAGAAG	CDS
*Atp5k*	86	TCCGCTCTGATCATCGGCATGG	CGCTGCTATTCTCCTCTCCTCCTC	CDS
*Ppa2*	85	TGAAGAAGTGTGACAAGGGAGCCA	GTGCAGTGGAAAGGGCTATCGC	CDS
*Ttn (pan)*	85	ACCAGAGCAAAGTGTAACGGCCA	TTCCCAAGCTATCCATGCACTCTGT	3′UTR
*CamK2d*	110	CAGCTACCGGACGGGATGTT	CTGACGGATCAGCTGAACTTGG	3′UTR

## Data Availability

The data presented in this study are available on request from the corresponding author. The presented gene array data are available in the supplementary material.

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
