# Peer review of "Pathophysiological Mechanisms of Cardiac Dysfunction in Transgenic Mice with Viral Myocarditis"

_cells, 2023, doi:10.3390/cells12040550_

Round 1

Reviewer 1 Report

In this paper, the author explored the virus-mediated cardiac pathophysiological processes in vivo and in vitro by using a transgenic mouse strain (TG) containing a CVB3ΔVP0 genome. The results found that in this viral myocarditis mouse model, mouse cardiac function, ECG, calcium homeostasis, and gene expression were significantly changed. The mechanisms of cardiac dysfunction may be caused by mitochondrial structure and gene expression alterations. Although the concept is novel, there is no sufficient data to support that mitochondrial impairment potentially contributes to cardiac contractile dysfunction and mechanistic insights are limited.

Major issue:

1.     On page 4, section “Ultrastructural analyses suggest mitochondrial degradation as a result of altered mitochondrial gene expression”, the results cannot be found in Figure 2. But I found the result in Figure 5. If it is Figure 5, please describe all the figures in Figure 5.

2.      For the gene array, please show the heatmap of the gene expression pattern.

3.      In the supplementary Figure 1, please explain why CVB3VP1 is positive in CVB3VP0 mouse cardiomyocytes and the purpose of the other markers staining.

Minor issue:

1.      Please check the Figure numbers in the manuscript. For example, on page 3, section “Marked alterations in ECGs of CVB3 mice”, there is no Figure 1f-1h that can be found in Figure 1. Another example after Figure 1 should be Figure 2, but in the manuscript, it directly goes to Figure 4.

2.      There are many grammar mistakes in the manuscript. Please check carefully and correct them. 

3.      Please correct the format based on the journal requirement. Especially the format of the figures. Please label the figures in the same font size.

4.      Please add the scale bar for all cell images.

Author Response

Reviewer 1:

Open Review

English language and style

( ) English very difficult to understand/incomprehensible
( ) Extensive editing of English language and style required
(x) Moderate English changes required
( ) English language and style are fine/minor spell check required
( ) I don't feel qualified to judge about the English language and style

Yes

Can be improved

Must be improved

Not applicable

Does the introduction provide sufficient background and include all relevant references?

( )

(x)

( )

( )

Are all the cited references relevant to the research?

(x)

( )

( )

( )

Is the research design appropriate?

( )

( )

(x)

( )

Are the methods adequately described?

( )

(x)

( )

( )

Are the results clearly presented?

( )

( )

(x)

( )

Are the conclusions supported by the results?

( )

( )

(x)

( )

Comments and Suggestions for Authors

In this paper, the author explored the virus-mediated cardiac pathophysiological processes in vivo and in vitro by using a transgenic mouse strain (TG) containing a CVB3ΔVP0 genome. The results found that in this viral myocarditis mouse model, mouse cardiac function, ECG, calcium homeostasis, and gene expression were significantly changed. The mechanisms of cardiac dysfunction may be caused by mitochondrial structure and gene expression alterations. Although the concept is novel, there is no sufficient data to support that mitochondrial impairment potentially contributes to cardiac contractile dysfunction and mechanistic insights are limited.

Answer:

Dear Reviewer 1,

we would like to thank you for the constructive feedback, which surely us helped to improve the quality of our manuscript. Hopefully, we have satisfactorily addressed all the desired corrections, but we are also open to further improvements if necessary. The corrections made in the manuscript are explained below.

Major issue:

  1. On page 4, section “Ultrastructural analyses suggest mitochondrial degradation as a result of altered mitochondrial gene expression”, the results cannot be found in Figure 2. But I found the result in Figure 5. If it is Figure 5, please describe all the figures in Figure 5.
  • The point raised by reviewer 1 is absolutely correct. The data described on page 4 in the section “Ultrastructural analyses suggest mitochondrial degradation as a result of altered mitochondrial gene expression” belong to Figure 5. We have corrected our mistake, but also relocated the data quantifying the cell length and area into figure 3, where they make more sense in context of readability of the manuscript. All data shown are now described in the manuscript in chronological order.

  1. For the gene array, please show the heatmap of the gene expression pattern.
  • We thank reviewer 1 for this suggestion. A heatmap of the gene array data makes sense in context of data visualization for the reader. We have added a heatmap of the significantly altered genes in figure 6 (now figure 6a). Additionally, we will add the gene array raw data and a sorted table of the significantly altered genes as supplementary data to the manuscript file to guarantee that a complete set of the data is available to readers.

  1. In the supplementary Figure 1, please explain why CVB3VP1 is positive in CVB3VP0 mouse cardiomyocytes and the purpose of the other markers staining.
  • We gladly explain the results of the immunostaining experiments shown in supplementary figure 1. It is not surprising that CVB3VP1 can be detected in CVB3DVP0-expressing mice. The DVP0 mutation is located at the cleavage site of the CVB3VP0 precursor protein, which is proteolytically cleaved into the functional capsid proteins VP2 and VP4. This hinders the formation of functional CVB3 virus capsids and renders the CVB3DVP0 mutant non-infectious. The generation of functional CVB3VP1 protein is not disturbed by the DVP0 mutation, which makes the CVB3VP1 protein detectable by immunohistochemistry. The localization of the CVB3DVP0 mutation is shown in figure 1a in the manuscript.

The immunostainings of the ion channels KCNQ1, hERG1, Cav1.2a and RYR2 show no differences between wt and CVB3-expressing isolated cardiomyocytes. For better understanding, we added a quantification of the fluorescence signal intensities (supplementary figure 1). The purpose of the immunostainings was to verify that CVB3DVP0-expression doesn’t affect cellular KCNQ1, hERG1, Cav1.2a and RYR2 expression and distribution (and so probably function), which supports the observation that action potential duration is not affected by CVB3DVP0 under resting conditions as well (figure 3c).    

Minor issue:

  1. Please check the Figure numbers in the manuscript. For example, on page 3, section “Marked alterations in ECGs of CVB3 mice”, there is no Figure 1f-1h that can be found in Figure 1. Another example after Figure 1 should be Figure 2, but in the manuscript, it directly goes to Figure 4.
  • The figure numbers in the manuscript were corrected. The figures are now described in numerical order in the manuscript.

  1. There are many grammar mistakes in the manuscript. Please check carefully and correct them. 
  • The manuscript was carefully read and corrected. We hope to have minimized the grammar mistakes in the manuscript.

  1. Please correct the format based on the journal requirement. Especially the format of the figures. Please label the figures in the same font size.
  • The manuscript was reformatted and the figures optimized to ensure sufficient quality for publication.

  1. Please add the scale bar for all cell images.
  • A scale bar was added to every cell image (or image set).

Reviewer 2 Report

This paper looked into the effect of CVB3 transfection on the cardiomyocytes due to reduced mitochondria function and loss of calcium homeostasis . However, There are a couple of issues that needs to be addressed before considering for publication.

Major issues

1.       Fig. 1B and 1C: the authors mentioned several parameters that are not changed and normal. Please include these data in the supplementary to prove that even they are not significantly different.

2.       Fig.1D: it might be better to show where the E peak and A peaks are, and show the E/A ratio in bar graph with statistics.  

3.       Fig.1F,G,H cannot be found in the main figures.

4.       Supplementary fig.1: please add quantification for the IF images.

5.       Please switch the order of the figures so they are in a more readable and organized way (Fig.2 switch with fig.4, Fig. 6 with Fig.5; also considering combining Fig.2 and Fig.5 since they are talking about similar things)

6.       Fig.6: Please add some IF staining for mitochondrial markers such as TOM20 in cardiomyocytes isolated from WT and CVB3 mice to validate the general mitochondrial number and function alteration.

Minor issues:

1.       Please consider separate the introduction into several paragraphs. It is a huge chunk of word that is hard to read and see the inner logic.

Author Response

Reviever 2:

Open Review

English language and style

( ) English very difficult to understand/incomprehensible
( ) Extensive editing of English language and style required
( ) Moderate English changes required
(x) English language and style are fine/minor spell check required
( ) I don't feel qualified to judge about the English language and style

Yes

Can be improved

Must be improved

Not applicable

Does the introduction provide sufficient background and include all relevant references?

(x)

( )

( )

( )

Are all the cited references relevant to the research?

(x)

( )

( )

( )

Is the research design appropriate?

( )

( )

(x)

( )

Are the methods adequately described?

(x)

( )

( )

( )

Are the results clearly presented?

( )

( )

(x)

( )

Are the conclusions supported by the results?

( )

( )

( )

( )

Comments and Suggestions for Authors

This paper looked into the effect of CVB3 transfection on the cardiomyocytes due to reduced mitochondria function and loss of calcium homeostasis. However, there are a couple of issues that needs to be addressed before considering for publication.

Answer:

Dear Reviewer 2,

We would like to thank you for the careful evaluation of our manuscript. We appreciate your constructive feedback and hope we could address your concerns in a sufficient and acceptable way. Your feedback surely helped to improve the quality of our manuscript tremendously. Following we added a point to point response to your issues.

Major issues

  1. 1B and 1C: the authors mentioned several parameters that are not changed and normal. Please include these data in the supplementary to prove that even they are not significantly different.

-  We agree that showing the mentioned, unaltered parameters in the manuscript would be beneficial. Therefore, we added the analysis of the mentioned heart mass and the end-diastolic volumes of the left and right ventricles of wt and CVB-expressing mice (Supplementary figure 3). Supplementary figure 3 is now also mentioned in the manuscript in the section “CVB3 mice exert impaired cardiac performance”.

  1. Fig.1D: it might be better to show where the E peak and A peaks are, and show the E/A ratio in bar graph with statistics.

      - We absolutely agree with the reviewer. We added the labelling for the E peaks and A peaks in figure 1d, but we kept the bar graphs in figure 1e unchanged, as they show an important dataset of the MRT measurements.  

  1. 3. 1F, G, H cannot be found in the main figures.

      - Figure 1f was added to figure 1, as it was missing in the previous manuscript version. The figures 1g and 1h are not mentioned in the manuscript anymore. The previously mentioned data of figure 1g and 1h are found in figure 2 and are now correctly mentioned in the manuscript.

  1. Supplementary fig.1: please add quantification for the IF images.

      - We performed signal intensity quantification for the ion channel immunostainings shown in supplementary figure 1, as suggested by the reviewer. The results are illustrated as bar diagram showing no significant alteration in signal intensity.

  1. Please switch the order of the figures so they are in a more readable and organized way (Fig.2 switch with fig.4, Fig. 6 with Fig.5; also considering combining Fig.2 and Fig.5 since they are talking about similar things)

- We agree with the reviewer about changing the order of the figures to increase readability of the manuscript. The figure order and description in the manuscript was changed, so that figures are discussed in chronological order. The figures 5a, b, were combined with figure 2 as suggested by the reviewer and figure 5 was switched with figure 4. Also, a new supplementary figure 3 was added to the manuscript, which was suggested by another reviewer. We think that the current figure order is now in line with the description in the manuscript and we hope the reviewer agrees with these changes.

  1. Fig.6: Please add some IF staining for mitochondrial markers such as TOM20 in cardiomyocytes isolated from WT and CVB3 mice to validate the general mitochondrial number and function alteration.

      - We are sorry to disappoint the reviewer with this issue. Unfortunately, the CVB3DVP0-expressing mouse line, which was analysed in this manuscript is non-existent anymore and there are also no cardiac tissue samples left for further analysis. This issue was already discussed with the editor, who assured us after his initial manuscript inspection that this should be unproblematic. As shown in figure 5, a total mitochondrial area analysis was performed on electron microscopic images of wt and CVB3-expressing mouse cardiomyocytes to illustrate mitochondrial degeneration induced by CVB3 expression.    

Minor issues:

  1. Please consider separate the introduction into several paragraphs. It is a huge chunk of word that is hard to read and see the inner logic.

-The introduction was separated into several paragraphs as suggested by the reviewer, which indeed increased readability of the introduction.

Reviewer 3 Report

The authors studied the pathophysiological mechanism of CVB3 virus on cardiac dysfunction using transgenic mice. 

The authors used transgenic mice producing replication-defective CVB3 virus and performed a series of characterizations. However, this exact same mice line has been established and characterized in previous studies (ref 17, 25). Here in this study, minimal new insights were provided especially on a mechanistic level. Many results here are identical to previous reports. The reviewer is not sure about the novelty of this work in its current form.

The authors also failed to investigate many important aspects of the pathology which they mentioned in the introduction such as host factors, inflammation, and potential cell death (a previously proposed mechanism).

Author Response

Reviewer 3:

Open Review

English language and style

( ) English very difficult to understand/incomprehensible
( ) Extensive editing of English language and style required
( ) Moderate English changes required
(x) English language and style are fine/minor spell check required
( ) I don't feel qualified to judge about the English language and style

Yes

Can be improved

Must be improved

Not applicable

Does the introduction provide sufficient background and include all relevant references?

( )

(x)

( )

( )

Are all the cited references relevant to the research?

( )

(x)

( )

( )

Is the research design appropriate?

( )

( )

(x)

( )

Are the methods adequately described?

( )

(x)

( )

( )

Are the results clearly presented?

(x)

( )

( )

( )

Are the conclusions supported by the results?

( )

( )

( )

( )

Comments and Suggestions for Authors

The authors studied the pathophysiological mechanism of CVB3 virus on cardiac dysfunction using transgenic mice. 

The authors used transgenic mice producing replication-defective CVB3 virus and performed a series of characterizations. However, this exact same mice line has been established and characterized in previous studies (ref 17, 25). Here in this study, minimal new insights were provided especially on a mechanistic level. Many results here are identical to previous reports. The reviewer is not sure about the novelty of this work in its current form.

The authors also failed to investigate many important aspects of the pathology which they mentioned in the introduction such as host factors, inflammation, and potential cell death (a previously proposed mechanism).

Answer: Most of the here shown data have not been addressed in the original manuscript dating back 24 years when several of the here applied techniques were not even yet available. Therefore, we are convinced that the novel data very much justify publication. Using the novel inducible transgenic human iPSC-model (Peischard et al., 2020) and trans-differentiation of these iPSCs to cardiac myocytes would allow for a very controlled cell population in an isogenic human cardiomyocytes model. According to this approach, we just used the system to study effects of the identical transgenic construct (CVB3ΔVP0) of pace maker channels (Peischard et al., 2022). Indeed, we plan to use the inducible CVB3ΔVP0-iPSC system to generate ventricular-like cardiomyocytes and study effects of the viral genome on mitochondria. As we have just started we can confirm at least that there similar morphological effects on mitochondria in these ventricular-like cardiomyocytes. However, it will take some more time until we will have generated a full set of data on this issue.

Literature:

Peischard S, Ho HT, Piccini I, Strutz-Seebohm N, Röpke A, Liashkovich I, Gosain H, Rieger B, Klingel K, Eggers B, Marcus K, Linke WA, Müller FU, Ludwig S, Greber B, Busch K, Seebohm G. The first versatile human iPSC-based model of ectopic virus induction allows new insights in RNA-virus disease. Sci Rep. 2020 Oct 8;10(1):16804. doi: 10.1038/s41598-020-72966-9.

Peischard S, Möller M, Disse P, Ho HT, Verkerk AO, Strutz-Seebohm N, Budde T, Meuth SG, Schweizer PA, Morris S, Mücher L, Eisner V, Thomas D, Klingel K, Busch K, Seebohm G. Virus-induced inhibition of cardiac pacemaker channel HCN4 triggers bradycardia in human-induced stem cell system. Cell Mol Life Sci. 2022 Jul 21;79(8):440. doi: 10.1007/s00018-022-04435-7.

Round 2

Reviewer 1 Report

The author answered all my questions about this manuscript, and the modified manuscript can be accepted now.

Reviewer 2 Report

The author addressed all my comments and I think it is good for publication.